# Self-Attention in Colors:
# Another Take on Encoding Graph Structure in Transformers

**Romain Menegaux**                                        *romain.menegaux@inria.fr*
*Univ. Grenoble Alpes, Inria, CNRS, Grenoble INP, LJK*
*38000 Grenoble, France*

**Emmanuel Jehanno**                                       *emmanuel.jehanno@inria.fr*
*Univ. Grenoble Alpes, Inria, CNRS, Grenoble INP, LJK*
*38000 Grenoble, France*

**Margot Selosse**                                          *margot.selosse@gmail.com*
*work done while at Univ. Grenoble Alpes, Inria, CNRS, Grenoble INP, LJK*
*38000 Grenoble, France*

**Julien Mairal**                                            *julien.mairal@inria.fr*
*Univ. Grenoble Alpes, Inria, CNRS, Grenoble INP, LJK*
*38000 Grenoble, France*

**Reviewed on OpenReview:** *https://openreview.net/forum?id=3dQCNqqv2d*

## Abstract

We introduce a novel self-attention mechanism, which we call CSA (Chromatic Self-Attention), which extends the notion of attention scores to attention *filters*, independently modulating the feature channels. We showcase CSA in a fully-attentional graph Transformer CGT (Chromatic Graph Transformer) which integrates both graph structural information and edge features, completely bypassing the need for local message-passing components. Our method flexibly encodes graph structure through node-node interactions, by enriching the original edge features with a relative positional encoding scheme. We propose a new scheme based on random walks that encodes both structural and positional information, and show how to incorporate higher-order topological information, such as rings in molecular graphs. Our approach achieves state-of-the-art results on the ZINC benchmark dataset, while providing a flexible framework for encoding graph structure and incorporating higher-order topology.

## 1  Introduction

The field of graph representation learning and graph-structured data has seen significant growth and maturity in recent years, with successful applications in a wide range of domains such as pharmaceutics, drug discovery (Gaudelet et al., 2021), recommender systems (Ying et al., 2018), or navigation systems (Derrow-Pinion et al., 2021). Key to these advances has been the success of graph neural networks (GNNs), which up until 2021 took mostly the form of local message-passing neural networks (MPNNs). In addition, the field has benefited from the development of a flurry of high-quality benchmark datasets (Dwivedi et al., 2020; Hu et al., 2020; 2021; Dwivedi et al., 2022b).

Local MPNNs have been known to suffer from limitations such as over-smoothing (Oono & Suzuki, 2020), bottlenecks in information propagation (Alon & Yahav, 2021) and limited expressiveness (Xu et al., 2019; Morris et al., 2021). This issue is alleviated by (i) adding positional or structural encodings to node features (Bouritsas et al., 2020; Abboud et al., 2021; Dwivedi et al., 2022a; Eliasof et al., 2023) and (ii) softly decoupling the message passing flow from the graph structure, by enhancing the graph with virtual nodes or higher order

structures – meaningful ones such as rings for molecules (Fey et al., 2020; Bodnar et al., 2021a), or simply random subgraphs (Zhao et al., 2022), leading to forms of *hierarchical* message-passing.

Another way to overcome these limitations is a whole avenue of work on graph transformers (GT) (Ying et al., 2021; Mialon et al., 2021; Kreuzer et al., 2021; Park et al., 2022; Hussain et al., 2022; Rampášek et al., 2022). Global attention GTs perform *dense* message-passing, where all nodes communicate with each other. Recently these have surpassed MPNNs on large-scale datasets (Ying et al., 2021; Park et al., 2022; Hussain et al., 2022; Rampášek et al., 2022). However they are not yet competitive on smaller datasets and tasks that are highly dependent on local substructures (Kreuzer et al., 2021; Rampášek et al., 2022). In this work, we develop a novel self-attention mechanism which – when coupled with expressive node interaction features – are competitive on benchmark datasets of all scales.

The output of the standard Transformer self-attention mechanism (Vaswani et al., 2017) is invariant to permutations. This issue is mostly solved for sequences by concatenating *positional encodings* to the element features. The natural equivalent for graphs are positional encodings based on the eigenvectors of the graph Laplacian (Dwivedi et al., 2020; Kreuzer et al., 2021). Albeit theoretically justified, in practice they are outperformed by other structural encodings specially tailored for graphs (Dwivedi et al., 2022a). Furthermore, even state-of-the-art node-wise positional encodings alone are not enough to bridge the performance gap between GTs and MPNNs (Rampášek et al., 2022). Furthermore *edge features* – which describe *semantic* relationships between nodes, such as the type of chemical bond in molecules – can be essential for valid graph representations, yet there is no standard or clear-cut way to incorporate them in GTs. In this work we give an answer the question:

*How to expressively incorporate both graph structural information and edge features in graph Transformers?*

Some recent GTs delegate this task to a local MPNN module (Wu et al., 2021; Chen et al., 2022; Rampášek et al., 2022), but otherwise all standalone GTs directly bias the attention matrix with *relative positional encodings* (Shaw et al., 2018) of different natures, engineering in both edge features and the adjacency matrix. We propose a unifying framework for the self-attention layer that flexibly subsumes most of the GT-flavors aforementioned.

**Contributions** :

- We enrich this framework by introducing Chromatic (or Channel-wise) Self-Attention (`CSA`): selectively modulating message channels, damping or highlighting them based on the relative positions of two nodes.

- A novel relative positional encoding scheme for graphs, based on random walks. We inject them in `CSA` and reach state-of-the-art results on several graph benchmarks.

- We showcase the generality of our framework by seamlessly integrating higher-order graph topological features such as rings. This augmented version `CSA`-rings greatly improves the state-of-the-art on the ZINC molecular dataset.

We show in rigorous ablation study on a set of graph benchmarks the added benefits of our proposed methods.

## 2  Setting and Related Work

### 2.1  Graph neural networks

Regular graph neural networks (GNNs) consider a graph as a set of node features $X \in \mathbb{R}^{N_{\mathrm{nodes}} \times d}$, an adjacency matrix $A \in \{0; 1\}^{N_{\mathrm{nodes}} \times N_{\mathrm{nodes}}}$ and (optionally) a set of edge features $E \in \mathbb{R}^{N_{\mathrm{edges}} \times d}$. The information contained in the adjacency matrix $A$ describes the connections between nodes and is henceforth referred to as the graph *structure*.

GNNs node representations $h$ (initialized by $h^{(0)} = X$) are iteratively updated in layers, by receiving and aggregating messages $M$ from other nodes. The edge features can also be updated, which we do not write

here.

$$h_i^{(\ell+1)} \leftarrow \mathrm{AGG}_{j \in \mathcal{N}(i)} \left\{ \mathrm{M} \left( h_j^{(\ell)}, e_{ij} \right) \right\}. \tag{1}$$

On the one hand, in local message passing networks (local-MPNNs) the aggregation spans only the neighbors $j$ of node $i$. The types of local-MPNNs – (Kipf & Welling, 2017; Bresson & Laurent, 2017; Corso et al., 2020) to cite just a few of the most popular variants – mostly differ on how the aggregation is performed. On the other hand, Graph Transformers (GTs), which can be interpreted as dense MPNNs, aggregate messages from all nodes together.

## 2.2 Transformer Architecture

In vanilla transformers, messages are linear transformations of node representations – the node *values $V$* –, and the aggregation is a simple weighted sum:

$$h_i \leftarrow h_i + \sum_{j=1}^{N_{\mathrm{nodes}}} \tilde{a}(i,j) \, V_j. \tag{2}$$

The coefficients in this weighted sum, the attention scores $\tilde{a}(i,j) \in [0,1]$, are obtained by a softmax normalization such that $\tilde{a}(i,j) = a(i,j)/\sum_k a(i,k)$ with

$$a(i,j) = \exp\left( \frac{1}{\sqrt{d}} Q_i \cdot K_j \right). \tag{3}$$

From now on, we omit the normalization term $\frac{1}{\sqrt{d}}$. $Q = W_Q h$; $K = W_K h$; $V = W_V h$ are commonly named the *query*, *key* and *value* matrices, respectively. In the multi-head setting, $N_{\mathrm{heads}}$ separated attention scores are computed, modulating the values $V$ by blocks of size $d/N_{\mathrm{heads}}$.

In this basic formulation, the update rule is invariant to permutations in the input nodes, and in fact oblivious to the graph structure and edge features $E_{ij}$. It is therefore crucial to include these, either by encoding and injecting them in the node features, or by tweaking the update rule.

## 2.3 Encoding graph structure

In this section, we give a brief overview of existing methods to encode graph structure and edge features in GTs.

**Local-MPNN hybrids**  One natural way to include the graph structure is simply to mix in local-MPNN layers, by either interleaving (Rampášek et al., 2022) or stacking (Wu et al., 2021; Chen et al., 2022) them with dense Transformer layers. The graph structural information, as well as the edge features, are handled by the local components, conveniently leveraging the rich literature on local MPNNs. In this work however we restrict ourselves to *purely attentional* networks, for which appropriate *positional encodings* (PE) are needed. We briefly go over the different flavors of graph PEs in the literature, see (Rampášek et al., 2022) for a more systematic overview and a thorough categorization. We adopt their distinction of *structural* and *positional* encodings.

**Node structural encodings (SE)**  Most successful node encodings for graph rely on local structural information, such as centrality measures (Ying et al., 2021) or local substructure counts (Bouritsas et al., 2020). One such encoding, related to the one we will introduce in section 3 – node-RWSE (Dwivedi et al., 2022a) – is based on random walks. Instrumental to the success of GTs, they have also shown to improve performance and expressivity in local MPNNs. However they cannot fully convey the relative *spatial* information in a Transformer.

For the sake of intuition, take the key and query matrices to be identity, and suppose we have concatenated a positional encoding $p$ to the features $h \leftarrow h \parallel p$. The attention score can then be written as the product of a semantic part and a positional one:

$$a(i,j) = \exp\left( h_i \cdot h_j \right) \exp\left( p_i \cdot p_j \right).$$

The original cosine positional embeddings $p$ designed for sequences have the desirable property that $p_i \cdot p_j$ is a function of the relative distance between $i$ and $j$: $|i - j|$. For most of the successful graph node encodings presented, these inner products can at most represent structural similarity i.e. how the local neighborhoods of nodes $i$ and $j$ compare to each other. They are independent of the distance between $i$ and $j$ within the graph, whereas the model could benefit from this information, for instance by attenuating signals coming from far away nodes.

**Node positional encodings (PE)** This distance-encoding role could be fulfilled by spectral positional encodings based on the eigenvectors of the graph Laplacian. These have been adapted into a node-PE (Dwivedi et al., 2020; Kreuzer et al., 2021), but suffer from implementation difficulties such as their invariance to sign flip of the eigenvectors, in practice hindering their performance (Dwivedi et al., 2022a). SignNet (Lim et al., 2022) propose to learn these encodings via a sign-invariant GNN.

Regardless of their particular type, current node-wise positional encodings are a helpful addition but do not suffice on their own to make global attention even remotely competitive with sparse message-passing (Rampášek et al., 2022).

**Relative positional encodings (RPE)** Each flavor of graph transformer comes with its own way of further incorporating the graph structure. One of the first graph transformers, SAN (Kreuzer et al., 2021) learns two different sets of attention parameters: one for adjacent pairs, the other for non-connected nodes. Graphormer (Ying et al., 2021) adds a learnable bias $b_{\phi(i,j)}$ to the attention score $a(i,j) = \exp\left(Q_i \cdot K_j + b_{\phi(i,j)}\right)$, indexed on the shortest path length $\phi(i,j)$ between $i$ and $j$. GraphiT (Mialon et al., 2021) bias the attention matrix with a graph kernel $K_r$ encoding the structural similarity: $a(i,j) = \exp\left(Q_i \cdot K_j\right) K_r(i,j)$. The choice of the kernel $K_r$ correspond to imposing different inductive biases. For instance $K_r(i,j) = \exp\left(p_i \cdot p_j\right)$ – with $p$ one of the structural encodings mentioned above – would encode structural similarity. On the other hand choosing the heat diffusion or the PageRank kernel as $K_r$ would encode a notion of walking distance between nodes. If $K_r$ is chosen as the adjacency matrix or the 1-step random walk then GraphiT becomes a local-MPNN, very similar to GAT (Veličković et al., 2018).

What we propose is an extension of GraphiT, and a generalization of Graphormer, where the attention mask $K_r$ is learned based on structural edge encodings.

## 3 Method

### 3.1 Unifying framework

Most of the self-attention modules of the aforementioned GTs can be rewritten as in equation 4, with $e_{ij}$ a scalar encoding the relationship between nodes $i$ and $j$.

$$a(i,j) = \exp\left(Q_i \cdot K_j + e_{ij}\right) \quad \in \mathbb{R}, \tag{4}$$

By choosing $e_{ij} = b_{\phi(i,j)}$ we recover Graphormer (the expression for $e_{ij}$ is slightly more complex in the presence of bond features, we do not write it here), and with $e_{ij} = \log(K_r(i,j))$ we replicate GraphiT. In another recent work – GRPE (Park et al., 2022) – re-use the query and key vector to compute the RPE bias $e_{ij} = Q_i \cdot E_{ij}^Q + K_j \cdot E_{ij}^K$.

Note that if $e_{ij}$ can be factorized as an inner product $p_i \cdot p_j$ (or more exactly as $W_Q p_i \cdot W_K p_j$), we recover the standard Transformer with positional encodings $p_i$.

To the best of our knowledge, this framework encompasses all previously proposed GTs, with the exception of EGT (Hussain et al., 2022), who scale the attention score *after* the $\ell_1$-normalization $\tilde{a}(i,j)$ with a log-normalized gating term on edge features $\sigma(E_{ij}^{\text{gate}}) \log(1 + \sum_k \sigma(E_{ik}^{\text{gate}}))$. Another recent GT (Kim et al., 2022) falls out of the scope of this framework entirely, by treating both nodes and edges as separate tokens.

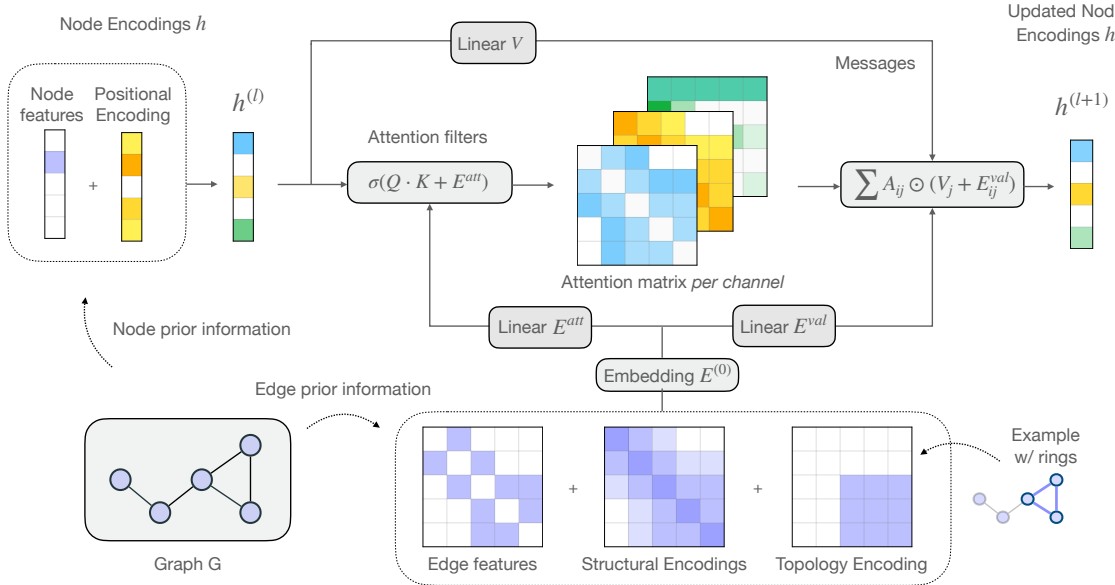

Figure 1: Chromatic Graph Transformer. (a) Graph features are preprocessed into node features $h$ (top-left) and two edge feature matrices $E^{att}$ and $E^{val} \in \mathbb{R}^{N_{\text{nodes}} \times N_{\text{nodes}} \times d}$ (bottom). (b) These are input to the CSA layers, which iteratively update the node representations $h$. (c) These representations are fed to a classification or regression head (not shown here).

**Monochrome attention**   In the coarsest form of our proposed model, `CSA`-mono, $e_{ij}$ is simply a scalar projection of the initial edge features $e_{ij} = W_E \cdot E_{ij}^{(0)}$. This is the "black and white", or mono-chromatic version of `CSA`.

## 3.2   Chromatic Self-Attention

**Attention filters**   In the self-attention formulation given above by (4 and 2), the *entirety of the node-node interaction* – both their structural and semantic relationship – is contracted into a *single* scalar $a(i, j)$ (resp. $N_{\text{heads}}$ scalars in multi-headed attention) which then modulates the node message $V_j$.

We refine this scheme by making the attention scores multi-dimensional, allowing a separate attention score $a_c(i, j)$ per output channel $c \in (1, d)$, generalizing the scalar attention score into an *attention filter* $\mathbf{a}(i, j) \in \mathbb{R}^d$.

$$\mathbf{a}(i, j) = \exp\left(Q_i \cdot K_j + E_{ij}\right) \quad \in \mathbb{R}^d \tag{5}$$

$\ell_1$-normalization is then performed across each channel

$$\bar{\mathbf{a}}(i, j) = \mathbf{a}(i, j) / \sum_k \mathbf{a}(i, k)$$

These filters are applied to the messages $V$ via an element-wise product.

$$h_i \leftarrow h_i + \sum_{j=1}^{N_{\text{nodes}}} \bar{\mathbf{a}}(i, j) \odot V_j \tag{6}$$

This allows to selectively attend to, or dampen certain channels in the incoming messages. Note that the two formulations (4) and (5) are equivalent if we impose $E_{ij}$ to be a constant vector, or a piecewise-constant vector in the multi-head setting – constant on the channels of each head.

We emphasize that (5) comes as an enhancement over standard multi-headed attention, and is notably *different* from using a multi-head version of (4) with $n_{\text{heads}} = n_{\text{channels}}$. This is explained in detail in Appendix B.

**Enhancing messages**  Most MPNNs that handle edge features combine them with the node features to form the message $M(h_j, E_{ij})$, with $M$ being a simple function such as concatenation, addition or elementwise product (Hu* et al., 2020; Corso et al., 2020). Similarly, we also enrich the individual node messages with relational information, by adding an edge *value* vector $E_{ij}^{\text{val}}$ to the node value vector $V_j$, the edge values being a linear transformation of the edge features $E_{ij}^{\text{val}} = W_{EV} E_{ij}^{(0)}$:

$$h_i \leftarrow h_i + \sum_{j=1}^{N_{\text{nodes}}} \tilde{\mathbf{a}}(i, j) \odot (V_j + E_{ij}^{\text{val}}) \tag{7}$$

Note that the update rule 7 opens up other possibilities for the attention filters, such as additive attention (Bahdanau et al., 2015). $\mathbf{a}(i, j) = \exp(Q_i + K_j + E_{ij})$ instead, completely foregoing the need for separate attention heads.

### 3.3   Design of the edge features

Fundamental to the success and expressive power of this scheme is the choice of these initial edge features.

**Semantic edge features**  After transforming the original or *semantic* edge features into suitable vector form $E^{\text{bond}} \in \mathbb{R}^{N_{\text{edges}} \times d}$, we extend them to node-pairs that were not originally connected in $A$, by introducing two learned embeddings $\in \mathbb{R}^d$: one for self-connections $E^{\text{self}}$ (if not already given), the other for non-connected nodes $E^{\text{n-c}}$.

$$E_{ij}^{\text{bond}} = \begin{cases} E_{ij}^{\text{bond}} & \text{if } i \text{ and } j \text{ are connected} \\ E^{\text{self}} & \text{if } i = j \\ E^{\text{n-c}} & \text{otherwise} \end{cases} \tag{8}$$

To encode both semantic and structural information in the final edge features $E$ we concatenate them with a relative positional encoding $E^{\text{RPE}}$:

$$E_{ij}^{(0)} = E_{ij}^{\text{bond}} \parallel E_{ij}^{\text{RPE}}. \tag{9}$$

Here we choose concatenation but any simple function such as addition or multiplication should yield similar results.

This system is general and flexible enough to leverage all possible pairwise relations in graphs that can be encoded onto vectors.

**Choice of the relative positional encoding**  For the RPE $E_{ij}^{\text{RPE}}$, we draw upon the family of functions of the successive powers of the random walk matrix $\text{RW} = D^{-1}A$ (where $D$ is the diagonal *degree* matrix):

$$E_{ij}^{\text{RPE}} = \Phi\left(\text{RW}_{ij}^1, \ldots, \text{RW}_{ij}^p\right) \quad \in \mathbb{R}^p, \tag{10}$$

With an appropriate choice of $\Phi$, these constitute powerful descriptors of the graph relations (Li et al., 2020). Indeed, one can recover positional information – the index of the first non-zero component is the shortest path distance (SPD) – but also structural information. For instance $\text{RW}_{ij}^1$ is nonzero iff $i$ and $j$ are neighbors and, in the absence of self-connections, $\text{RW}_{ii}^3$ is nonzero iff $i$ is part of a triangle.

We give two possible RPEs constructed from these features. One of them – SPDE – is present in some form in other GTs (Ying et al., 2021; Park et al., 2022; Hussain et al., 2022). The other – RWSE – is commonly used as a node-wise SE but has to our knowledge never been used before as an RPE.

**Shortest Path Distance Encoding (SPDE)**  The shortest path distance encoding is obtained by choosing $\Phi$ as a $d$-dimensional embedding of the index of the first nonzero component:

$$E_{ij}^{\text{SPDE}} = W_{\text{SPDE}} e_{\phi(i,j)} \tag{11}$$

with $W_{\text{SPDE}} \in \mathbb{R}^{d \times p}$ the embedding vectors and $e_{\phi(i,j)} \in \mathbb{R}^p$ the one-hot encoding of the shortest path length $\phi(i,j)$. Similarly to the semantic features above, we complete the feature matrix with a special embedding for nodes unreachable in less than $p$ hops, and another one for self connections.

**Random Walk Structural Encoding (RWSE)**
In our proposed RPE, $\Phi$ is simply a matrix multiplication with learnable weights $W_E^{\text{RWSE}}$:

$$E_{ij}^{\text{RWSE}} = W_E^{\text{RWSE}} \cdot \left[ \text{RW}_{ij}^1, \dots, \text{RW}_{ij}^p \right] \qquad (12)$$

If the weights matrix $W_E^{\text{RWSE}}$ is constrained to be non-negative, these features can be considered as re-weighted diffusion kernels between nodes, hence constituting a learnable and multi-dimensional ex-

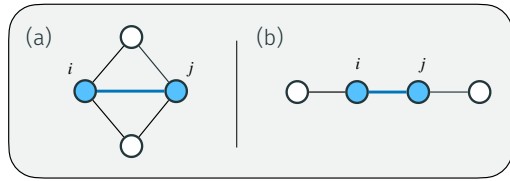

Figure 2: Shortest Path Distance Encoding cannot distinguish the relationship $i \leftrightarrow j$ in the two above graphlets, as they are both neighbors with $\phi(i,j) = 1$. On the other hand the three first coordinates of the RWSE$_{ij}$ vectors are markedly different: $[0.33, 0.33, 0.26]$ and $[0.5, 0.0, 0.63]$ for the graphlets on the left and right, respectively.

tension of GraphiT. We argue that RWSE is a richer RPE than SPDE as it also preserves some local structural information. A simple illustrative example is shown in Figure 2.

### 3.4 Adding substructures

Incorporating higher-order structure information can be beneficial to performance. In molecular graphs for example, atoms in a *ring* (a chordless cycle) share covalent electrons, which influence their distribution in space and their chemical properties. Hierarchical message-passing schemes (Fey et al., 2020; Bodnar et al., 2021b;a) were designed both to propagate information effectively along meaningful substructures and to suitably modulate the messages. We emulate this form of hierarchical message-passing in `CSA` by injecting specific topological information directly into the edge features (and hence in both the attention matrix and the messages).

For a chosen set of structures (graphs) $\mathcal{S}$ we define a binary relation $\underset{\mathcal{S}}{\approx}$ on nodes of a graph in definition 3.1. $\underset{\mathcal{S}}{\approx}$ is symmetric but not necessarily reflexive or transitive.

**Definition 3.1.** Given a set of structures $\mathcal{S}$, two nodes $i$ and $j$ of a graph $G$ are $\mathcal{S}$-*bound* – written $i \underset{\mathcal{S}}{\approx} j$ – if and only if there exists an element $S \in \mathcal{S}$ and a subgraph $\tilde{G}$ of $G$ containing both $i$ and $j$ such that $\tilde{G}$ is *isomorphic* to $S$.

$$
\begin{aligned}
i \underset{\mathcal{S}}{\approx} j &\iff \exists S \in \mathcal{S}, \tilde{G} \in \text{Sub}(G) \\
&\text{s.t.}\, i, j \in \tilde{G} \text{ and } S \sim \tilde{G}
\end{aligned}
\qquad (13)
$$

We then design boolean features based on this relation $e_{ij}^{\mathcal{S}} = \mathbb{1}(i \underset{\mathcal{S}}{\approx} j)$. In the more generic form of our topology encoding, which can be applied to any type of data, these features are embedded and then added to the edge features:

$$E_{ij} \leftarrow E_{ij} + \text{Embed}(e_{ij}^{\mathcal{S}})$$

In the cases where the original edge features $E^{\text{bond}}$ are *categorical*, i.e. belong to a discrete set $C$, one can give more degrees of freedom to the substructure encoding by forming the Cartesian product of the semantic and the topology features, effectively doubling the number of edge types.

$$E_{ij}^{\text{bond}} \leftarrow \text{Embed}\left( \left( E_{ij}^{\text{bond}}, e_{ij}^{\mathcal{S}} \right) \right)$$

Note this could also be done on the SPDE positional encoding for instance. Although this choice of encoding is less easily applicable (due to the restrictive condition on the edge features) it is also more expressive, and yields slightly improved results empirically.

In practice we have implemented this topology encoding for rings in molecular graphs, by choosing $\mathcal{S}$ as the set of rings up to a certain length, but the same could be done for any type of substructure deemed relevant to the task at hand.

### 3.5 Discussion and implementation

**Model architecture**  The main components of our architecture are illustrated in Figure 1. As is standard in Transformer layers, our `CSA` layer is composed of the self-attention block followed by 2 linear layers with batch normalization. We stack $L$ of these layers which yield final representations $h^{(L)}$. These are fed to a prediction head that depends on the task (graph/node/edge classification/regression).

$$h^{(l+1)} = \texttt{CSA}(h^{(l)}, E^{(0)})$$
$$E^{(0)} = \texttt{EdgeEncoder}(E^{\text{bond}}, A)$$
$$h^{(0)} = \texttt{NodeEncoder}(X, A)$$

**Regularization**  Aside from standard regularization techniques, we perform dropout directly on the attention tensor, silencing either whole nodes (node-ablation), connections (edge-ablation) or channels (feature dropout). All of these are options in our model and our code.

**Generality**  Semantic edge feature information are preserved (this is guaranteed by adding it to the messages directly), and structural information is also taken into account, modulating both the attention matrix and the messages.

Note that this framework is general enough to be applied to other types of data such as sequences, images or 3D point clouds. The graph structure is not essential to the `CSA` formulation, which can be transposed to other fields by designing an appropriate RPE – for instance embedding 3D distance and angles in 3D points clouds, or pixel distances in images.

**Limitations**  There are several limitations with our proposed model, the main one being the *scalability* to large and sparse graphs. Indeed the computational complexity of our method scales as $O(|N_{\text{nodes}}|^2 d)$, compared to the $O(|N_{\text{edges}}| d)$-complexity of local MPNNs. For small graphs however this is partially offset by the efficiency of GPU dense matrix computation. To scale our method to larger graphs one could explore the numerous attempts at a linear or sub-quadratic Transformers (Zaheer et al., 2020; Jaegle et al., 2022), transposing them to our framework. The pre-processing steps also incur a one-time cost of $O(|N_{\text{nodes}}|^3 p)$ for the computation of the RWSE or the SPDE base values.

Another limitation could be the reliance on hand-crafted features in the RPEs – although the same could be said of all GTs – but in fact there are very few hyper-parameters to tune, and fixed step sizes of RWSE-16, SPDE-8 work across a variety of datasets.

**Implementation**  We integrate our proposed `CSA` layer in the general GraphGPS framework, which enables testing on several benchmark datasets, choice of a wide range of positional encodings as well as adding local MPNN layers in parallel (we do not use this feature in this paper though). We also add our implementations of the relative positional encodings Edge-RWSE and SPDE. The code of the model, and scripts to reproduce the experiments are freely available at `https://github.com/inria-thoth/csa`.

## 4 Experiments

### 4.1 Experimental setup

We evaluate the performance of our model on 3 medium-scale benchmark datasets from (Dwivedi et al., 2020): ZINC, PATTERN and CLUSTER and on the large molecular dataset PCQM4Mv2 (Hu et al., 2021). We arrange the compared methods in 3 broad categories: MPNNs, extended (or hierarchical) MPNNs, and graph transformers. We chose to include the best performing ones (to our knowledge) in each category, with an emphasis on the more comparable graph transformers. We show SOTA results in all 4 datasets.

We conduct ablation studies to showcase the individual contributions of (i) chromatic attention, (ii) attending to distant nodes (iii) the choice of the RPE and (iv) incorporating topological sub-structures.

| | Model | ZINC
MAE ↓ |
|---|---|---|
| **MPNN** | GCN (Kipf & Welling, 2017) | $0.367 \pm 0.011$ |
| | GatedGCN (Dwivedi et al., 2022a) | $0.090 \pm 0.001$ |
| | GPS (Rampášek et al., 2022) | $0.070 \pm 0.004$ |
| **h-MPNN** | CIN (Bodnar et al., 2021a) | $0.079 \pm 0.006$ |
| | CRaWL (Toenshoff et al., 2021) | $0.085 \pm 0.004$ |
| | GIN-AK+ (Zhao et al., 2022) | $0.080 \pm 0.001$ |
| **Transformers** | SAN (Kreuzer et al., 2021) | $0.139 \pm 0.006$ |
| | Graphormer (Ying et al., 2021) | $0.122 \pm 0.006$ |
| | SAT (Chen et al., 2022) | $0.094 \pm 0.008$ |
| | EGT (Hussain et al., 2022) | $0.108 \pm 0.009$ |
| | GRPE (Park et al., 2022) | $0.094 \pm 0.002$ |
| | `CSA` (ours) | $\mathbf{0.070 \pm 0.003}$ |
| | `CSA`-rings (ours) | $\mathbf{0.056 \pm 0.002}$ |

Table 1: Results on the ZINC dataset, ordered by the broad model class and performance. (GPS is placed among local MPNNs here, as their classification performance can be fully imputed to their local module).

Following (Rampášek et al., 2022), the ablation studies are performed on ZINC and a subset of PCQM4Mv2 containing 10% of the whole training set (but the same validation set as the full version).

Little hyperparameter search was done, defaulting to the setup in GraphGPS or other comparable models whenever possible. We chose maximum ring-size as in (Bodnar et al., 2021a) $k = 18$ for ZINC, and $k = 6$ for OGB datasets. For ZINC we use the categorical ring encoding scheme described in section 3.4 and for OGB datasets we use the boolean encoding.

All models were trained on a single NVidia V100 GPU system, with 16GB or 32GB memory depending on the dataset. See Appendix A for the complete experimental details and hyperparameters.

### 4.2 Benchmarking `CSA`

**ZINC and Benchmarking Graph Neural Networks** ZINC is a popular molecular graph regression benchmark, for which Graph Transformers have historically lagged behind their sparse counterparts. The dataset splits are fixed as in (Dwivedi et al., 2020), with $10K$ graphs in the training set, $1K$ in the both the validation and test sets. The graphs contain on average $\sim$ 20-30 nodes.

Our results are presented in Table 1, in which the ring-augmented version of our model `CSA`-ring improves the state of the art by a significant margin. Adding ring information greatly improves the performance in ZINC, which is to be expected as the artificial objective partially depends on the counts of these patterns.

However we note that `CSA` *achieves state-of-the-art results even without this substructure information*, far surpassing other standalone GTs.

**PATTERN and CLUSTER** PATTERN and CLUSTER are synthetic datasets of inductive *node-level classification*. In both cases, graphs were generated using a Stochastic Block Model (SBM). In

| Model | PATTERN
Accuracy ↑ | CLUSTER
Accuracy ↑ |
|---|---|---|
| SAN | $86.581 \pm 0.037$ | $76.691 \pm 0.65$ |
| Graphormer | − | − |
| SAT | $86.848 \pm 0.037$ | $77.856 \pm 0.104$ |
| EGT | $86.821 \pm 0.020$ | $\mathbf{79.232 \pm 0.348}$ |
| GPS | $86.685 \pm 0.059$ | $78.016 \pm 0.180$ |
| `CSA` (ours) | $\mathbf{87.011 \pm 0.036}$ | $\mathbf{79.175 \pm 0.126}$ |

Table 2: Results on the node classification datasets PATTERN and CLUSTER.

PATTERN, one must detect which nodes were generated with different SBM parameters than the rest of the graph, and in CLUSTER one must assign each node to one of the 6 clusters (or blocks) it belongs to. These datasets contain 14K and 12K graphs, each with 100 nodes and 2K edges on average (much denser than molecule graphs). As shown in Table 2, we reach state-of-the-art performances on both datasets.

**Larger scale datasets**  Finally, we assess the performance of our method on the large-scale benchmark PCQM4Mv2. The dataset is composed of 3.7M small undirected graphs (14.6 nodes on average).

| | Model | PCQM4Mv2 | |
|---|---|---|---|
| | | **Validation MAE** ↓ | **# Param.** |
| MPNN | GCN | 0.1379 | 2.0M |
| | GCN-virtual | 0.1153 | 4.9M |
| | GIN | 0.1195 | 3.8M |
| | GIN-virtual | 0.1083 | 6.7M |
| Transformers | Graphormer | 0.0864 | 48.3M |
| | EGT | 0.0869 | 89.3M |
| | GRPE | 0.0890 | 46.2M |
| | GPS-small | 0.0938 | 6.2M |
| | GPS-medium | 0.0858 | 19.4M |
| | `CSA`-small (ours) | 0.0898 | 2.8M |
| | `CSA`-deep (ours) | **0.0853** | 8.3M |

Table 3: Results on the PCQM4M-v2 dataset, ordered by the broad model class and performance. As the test set was kept private by challenge organizers, evaluation is done using the validation set.

We test both a *small* and a *deep* version of our model in Table 3, which outperforms SOTA with less parameters. The difference in performance is however not as significant as for the ZINC dataset, we presume this is because the comparative advantages of our method diminishes with larger models and datasets.

### 4.3 Ablation study

**Usefulness of the chromatic attention**  In Figure 3 we visualize the different learned attention score matrices, for 3 output channels of a 16-channel model trained on ZINC. Each of these channels clearly focuses on structurally different nodes: the first channel on neighboring nodes, the second one on atoms in the same ring, and the third one focuses exclusively on faraway nodes. The figure shows both the complete node-node attention maps for each channel, and also some selected rows plotted on the graphs, for ease of visualization. These show the attention scores of the highlighted node, across 4 structurally diverse nodes.

The attention maps for all 16 channels are available in the appendix.

To ensure that this is not a learned artifact and that the model actually benefits from these extra degrees of freedom, we empirically validate the increased expressiveness of `CSA` in Table 5 by successively ablating the chromatic nature of the attention scores, and the edge value vectors. On the two datasets we have tested, both additions make an impact.

| RPE | ZINC MAE ↓ | PCQM4Mv2 MAE ↓ |
|---|---|---|
| None | $0.112 \pm 0.002$ | $0.1192 \pm 0.0001$ |
| SPDE | $0.068 \pm 0.002$ | $0.1123 \pm 0.0006$ |
| RWSE | $0.066 \pm 0.001$ | $0.1109 \pm 0.0002$ |

Table 4: `CSA` performance with different relative positional and structural encodings (RPE).

**Structural encoding**  We confront our proposed positional and structural Random Walks Encodings with the Shortest Path Distance Encoding in Table 4 and confirm the intuition that RWSE outperforms SPDE, regardless of the topological encoding.

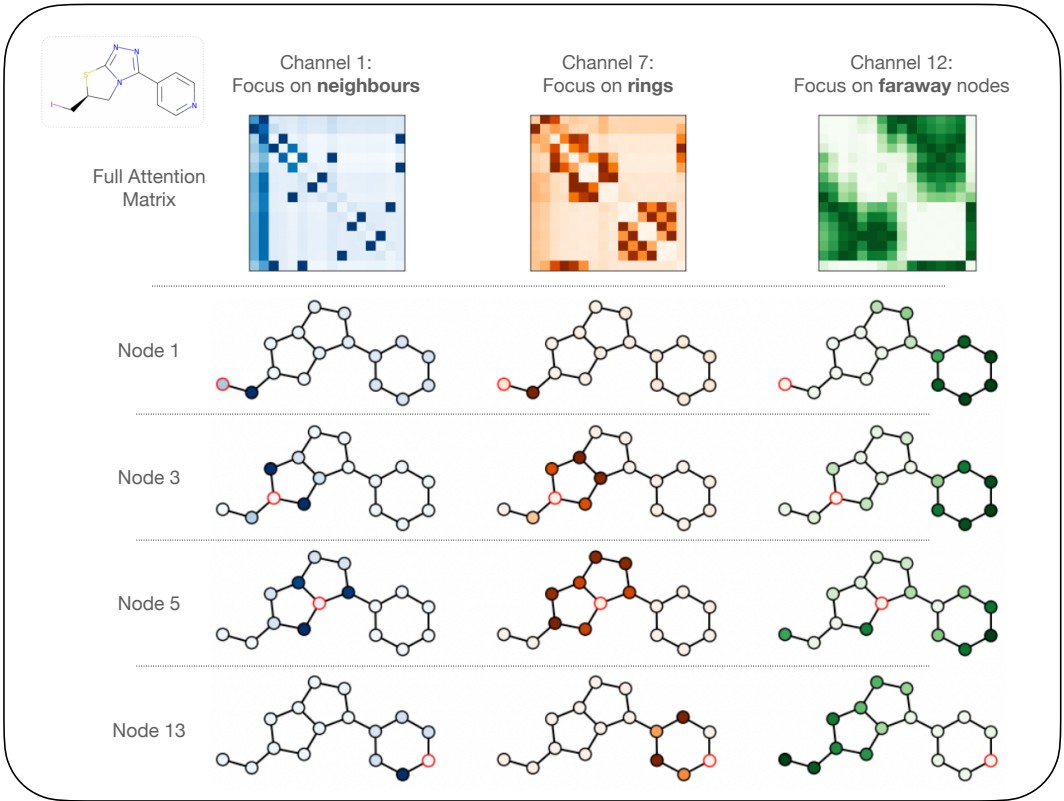

Figure 3: Attention scores for multiple nodes and channels at the last layer of a 16-channel model trained on ZINC. Scores are shown for the molecule `ZINC44549294`. Each column corresponds to a different channel, which we hand-picked based on their diversity and interpretability. The top row shows the attention channel scores for all node pairs, and each subsequent row corresponds to a different receiving node, highlighted in red.

| Color | Edge Value | ZINC | PCQM4Mv2 subset |
|:---:|:---:|:---:|:---:|
| | | MAE ↓ | MAE ↓ |
| − | − | $0.214 \pm 0.067$ | $0.129 \pm 0.0010$ |
| ✓ | − | $0.176 \pm 0.053$ | $0.121 \pm 0.0018$ |
| − | ✓ | $0.068 \pm 0.004$ | $0.112 \pm 0.0008$ |
| ✓ | ✓ | $0.066 \pm 0.001$ | $0.111 \pm 0.0002$ |

Table 5: Added value of including multidimensional attention filters ("Color" column) and injecting edge information in the *value* messages ("Edge Value" column)

## 5 Conclusion

We have proposed an extension of the Transformer self-attention, generalizing attention scores to attention filters. We adapt the resulting model to graph representation learning, further extending the self-attention mechanism by incorporating edge features in the message values. We combine our model with a novel relative positional encoding scheme – and a topology encoding one – and we show its added value on a diverse set of graph benchmarks.

## Acknowledgments

This project was supported by ANR 3IA MIAI@Grenoble Alpes (ANR-19-P3IA-0003) and by ERC grant number 101087696 (APHELEIA project). The bulk of the computations were performed using HPC resources from GENCI–IDRIS (Project-[A0131012608]).

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

## A    Experimental Details

**Implementation**   As previously mentioned, our code is integrated in the GraphGPS framework introduced by (Rampášek et al., 2022). In particular, the data-loaders and splitting strategies are kept the same to ensure a fair comparison with other methods. Results in the benchmark section were all obtained with 10 seeds each, except for PCQM4Mv2-full which was run on a single seed. Results in the ablation study are aggregated over 4 seeds. We report the mean loss along with its standard deviation over runs.

**Hyper-parameters**   The table 6 recaps the chosen hyperparameters for each dataset, as well as training execution time. In all our experiments we use the AdamW (Loshchilov & Hutter, 2019) optimizer with default values for $\beta_1$, $\beta_2$ and $\epsilon$, and a cosine learning rate scheduler with warmup.

For the ZINC ablation studies, the number of epochs was reduced from 2000 to 1000, and rings were not taken into account.

| Hyperparameter | ZINC | PATTERN | CLUSTER | PCQM4Mv2 CSA-*small* | PCQM4Mv2 CSA-*deep* |
|---|---|---|---|---|---|
| # `CSA` Layers | 10 | 6 | 16 | 5 | 16 |
| Hidden Dim | 64 | 64 | 48 | 256 | 256 |
| # Heads | 4 | 4 | 8 | 16 | 16 |
| Dropout | 0 | 0 | 0.1 | 0 | 0 |
| Attention Dropout | 0.5 | 0.5 | 0.5 | 0.5 | 0.5 |
| Node Positional Encoding | RWSE-20 | RWSE-16 | RWSE-16 | None | None |
| PE dim | 28 | 16 | 16 | - | - |
| Relative Positional Encoding | RWSE-20 | RWSE-16 | RWSE-16 | RWSE-16 | SPDE-8 |
| Edge parameter sharing | ✓ | ✗ | ✗ | ✓ | ✓ |
| Rings | ✓ | ✗ | ✗ | ✗ | ✓ |
| Max rings size | 18 | - | - | - | 6 |
| Encoding | categorical | - | - | - | additive |
| Batch Size | 32 | 32 | 16 | 256 | 256 |
| Learning rate | 0.001 | 0.0005 | 0.0005 | 0.0005 | 0.0001 |
| # Epochs | 2000 | 100 | 100 | 300 | 150 |
| # Warmup epochs | 50 | 5 | 5 | 10 | 10 |
| Weight decay | 1e-5 | 1e-5 | 1e-5 | 0 | 0 |
| # Parameters | 350k | 259k | 382k | 2.8M | 8.2M |
| Time (per epoch) | 21s | 40s | 85s | 822s | 1900s |

Table 6: Hyperparameters used for all datasets. Ablation studies on the PCQM4Mv2-subset are all done based on the `CSA`-*small* configuration.

| Hyperparameter | ZINC | PATTERN | CLUSTER | PCQM4Mv2 *small* model | PCQM4Mv2 *medium* model |
|---|---|---|---|---|---|
| GraphGPS (Time/epoch) | 21s | 32s | 86s | 619s | 1124s |
| `CSA` (Time/epoch) | 21s | 40s | 85s | 822s | 1520s |

Table 7: Training times of `CSA` on a single A100 GPU, compared to GraphGPS (Rampášek et al., 2022)

**Edge parameter sharing**    To reduce computation time and in some cases improve performance, we provide the option to share the edge representations $E^{att}$ and $E^{val}$ among all layers.

To get a frame of reference for the execution times, we report the training times of GraphGPS (Rampášek et al., 2022) in table 7.

In figure 4 we show all 16 channels of the first layer of a single attention head `CSA`model trained on ZINC. 3 of these were chosen in figure 3 of the main text. One can clearly see that channels are learning different attention patterns, with some being identifiable as immediate neighbors, nodes in the same ring, and also complementary channels concentrating on faraway nodes.

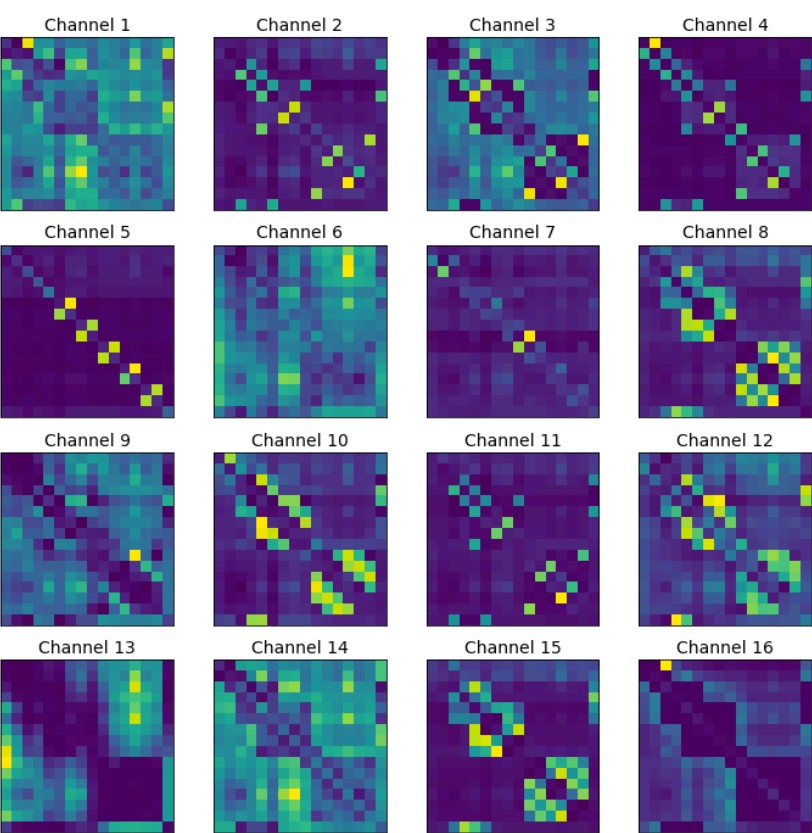

Figure 4: Attention scores for all 16 channels at the first layer of a single-head model trained on ZINC. Scores are shown for the molecule ZINC44549294.

| Color | Heads | ZINC | PCQM4Mv2 |
|:---:|:---:|:---:|:---:|
| − | 1 | $0.074 \pm 0.004$ | $0.147 \pm 0.003$ |
| ✓ | 1 | $0.0059 \pm 0.001$ | $0.121 \pm 0.001$ |
| − | best | $0.060 \pm 0.002$ | $0.1145 \pm 0.0002$ |
| ✓ | best | $\mathbf{0.056 \pm 0.001}$ | $\mathbf{0.112 \pm 0.001}$ |
| − | $h = d$ | $0.062 \pm 0.001$ | $0.118 \pm 0.005$ |

Table 8: Performance of CSA vs CSA-mono with different number of heads. best is 8 for 64 dimensions for ZINC, 16 for 256 dimensions in PCQM4Mv2. (when $h = d$ the color and monochrome versions are the same)

## B  Difference between CSA and multi-head attention

In short, our method *enhances* the standard multi-head attention (which we call CSA-mono in the paper).

Indeed, note $d$ the number of channels and $h$ the number of heads. For a given channel $c \in [1, d]$, we denote $d_h = d/h$ the implicit dimension of the attention heads, and $c_h = \lfloor c/d_h \rfloor$ the index of the attention head which channel $c$ belongs to. The $c$-th channel of the unnormalized attention score between two nodes $q$ and $k$ of dimension $d$, linked by the edge $e$ is then given by:

- **Standard multi-head attention** (called CSA-mono in the paper) with $d = h$:

$$a_c = \frac{1}{\sqrt{d_h}} \sum_{l=c_h d_h}^{c_h d_h + d_h - 1} q_l k_l + e_{c_h}$$

    The edge bias $e_{c_h}$ here is constant across all channels of the same attention head. (In fact in most other GTs this bias is constant across all attention heads:
    $e_{c_h} = e_1, \forall c$).

- **Colored multi-head attention** (CSA-color with $1 < h < d$), $e \in \mathbb{R}^d$:

$$a_c = \frac{1}{\sqrt{d_h}} \sum_{l=c_h d_h}^{c_h d_h + d_h - 1} q_l k_l + e_c$$

    The difference here is that the edge bias $e_c$ varies *per channel* instead of *per head.*

In multi-headed dot-product attention, choosing the hyperparameter $h = n_{heads}$ amounts to striking a balance between the *selectivity* of the per-head cosine similarities $Q_h \cdot K_h$ (of implicit dimension $d/h$), and the potential *diversity* of those similarities across attention heads ($h$ different scores).

In CSA we maintain this balancing option intact, but also keep a separate edge bias $e_c$ per channel, without loss of expressivity.

We compare CSA with its monochrome version in table 8, varying the number of attention heads.

