# OpenReview forum: "Self-Attention in Colors: Another Take on Encoding Graph Structure in Transformers"
_TMLR — Accepted by TMLR_

### Review · Reviewer_3Q4W · 2023-06-08

**Summary Of Contributions:**

This paper introduces a new framework for computing attention in graph transformers, by taking multidimensional attention filters instead of scalar attention coefficients. This allows for greater flexibility in choosing which part of a node's features the network passes to another node. The paper also introduces a new type of positional encoding based on the random walk adjacency matrix, which is used to assign learnable encodings to the edges.

**Audience:**

Yes

**Claims And Evidence:**

Yes

**Requested Changes:**

- It would be nice to have a comparison of the runtime of CSA with other architectures.

**Strengths And Weaknesses:**

## Strengths
- The paper is well-written and clear. The model is well-motivated with several examples to provide intuition, and there is a comprehensive comparison of the architecture with other types of graph transformers. The experimental setup is well-documented.
- The proposed chromatic self-attention generalizes existing graph transformer architectures and is more flexible, allowing the network to learn positional encodings and making fewer arbitrary choices.
- CSA appears be competitive with state-of-the-art graph transformers on several datasets.
## Weaknesses

- The proposed architecture has issues with scalability, since computing attention has quadratic complexity in the number of nodes in the graph. It is not clear how much of the benefit of this model is due to the encoding structure versus the type of attention used. Would this advantage carry over even using linear or sparse attention like other graph transformers?
- The new relative positional encoding introduced (RWSE) also seems to be expensive to compute - it requires computing $k$-th powers of the normalized adjacency matrix of the graph.

---

> ### Author Response · Authors · 2023-07-28
>
> Thank you for taking the time to review. We hope the following will answer your comments, and we are happy to discuss if you have further questions.
>
> > The proposed architecture has issues with scalability, since computing attention has quadratic complexity in the number of nodes in the graph. It is not clear how much of the benefit of this model is due to the encoding structure versus the type of attention used. Would this advantage carry over even using linear or sparse attention like other graph transformers?
>
> This is partly answered in the ablation study, where we show both the encoding schemes as well as the attention mechanism bring added performance. On the ZINC benchmark the main performance boosts come from adding ring information, as well as adding edge values $E^{val}$. We presume this advantage would also carry over to other models.
> Adding topology information such as rings has been shown help on ZINC for more generic MPNNs [Fey et al., 2020; Bodnar et al., 2021].
>
> The other graph transformers that are competitive on PCQM4Mv2 all use dense attention, not sparse. The gains on this large dataset from our encodings and our attention mechanism are minor compared to other GTs. Our intuition is that for larger models and datasets the more sophisticated mechanisms do not matter as much (for instance there is less need for the fine-grained colored attention when there are many attention heads).
>
> > The new relative positional encoding introduced (RWSE) also seems to be expensive to compute - it requires computing -th powers of the normalized adjacency matrix of the graph.
>
> These are computed only once during training, but even if computed at every pass they only correspond to $p$ $(N,N)$ matrix multiplications, where $p$ is the number of walking steps and $N$ the number of nodes in the graph. For small graphs this is rather small compared to the later matrix operations in the network.
>
> > It would be nice to have a comparison of the runtime of CSA with other architectures.
>
> We lack data from other graph transformers to perform an extensive comparison, but for similar-sized models from GraphGPS [Rampasek et al., 2022], and on similar hardware (a single A100 GPU) our method is just as fast on ZINC (20s/epoch), and around 30% slower on the other datasets (1500s/epoch vs 1100s/epoch for the *medium* models on PCQM4Mv2).
>
> We have added this as table 7 in the Appendix.

---

### Review · Reviewer_y6nv · 2023-06-18

**Summary Of Contributions:**

The paper presents CSA (Chromatic Self-Attention), a novel self-attention mechanism for encoding graph structure in transformers. The approach is showcased in CGT (Chromatic Graph Transformer), which integrates graph structural information and edge features without local message-passing. By incorporating edge features and relative positional encoding, the method achieves state-of-the-art results on the benchmark datasets with various scales. It provides a flexible framework for graph structure encoding and incorporates higher-order topology.

**Audience:**

Yes

**Broader Impact Concerns:**

I do not have further concerns on the ethical implications.

**Claims And Evidence:**

Yes

**Requested Changes:**

1.	Add some statistics of the used datasets to show they have different scales from small to large.
2.	Some equations are not indexed.
3.	Introduce some baselines and metrics. For example, why you use MAE for some datasets and use accuracy for others?
4.	Also refer to the weakness.

**Strengths And Weaknesses:**

Strengths :
1. A new CSA Mechanism extends the traditional attention scores to attention filters. This novel approach independently modulates feature channels, allowing for flexible encoding of graph structure through node-node interactions.
2. The proposed method, CGT incorporates both graph structural information and edge features without the need for local message-passing components. By incorporating edge features in the message values, the model captures rich information from the graph, enhancing its expressive power. It can be considered as the generalized version of GraphiT and Graphormer,
3.	State-of-the-Art Performance on the real-world and synthetic datasets of all scales, showing the effectiveness of the method on small and large datasets.

Weaknesses:
1.	Lack the analysis of complexity. Will the method cost more time or space compared with other graph transformer?
2.	Lack the introduction of some baselines and metrics. Why you select these methods as baselines? Are they representative?
3.	Code is not published.

---

> ### Author Response · Authors · 2023-07-28
>
> Thank you for taking the time to review our paper, we hope the following will answer your points.
>
> > 1. Lack the analysis of complexity. Will the method cost more time or space compared with other graph transformer.
>
> The complexity analysis is indeed lacking, we give one here, both in terms of FLOPs, time and space.
>
> Let $N$ be the number of nodes in the graph, $d$ the dimension of the hidden features $h$ (we suppose they are the same from layer to layer), and $h$ the number of attention heads.
>
> **FLOPs and time complexity**
>
> - The total number of FLOPs for a standard self-attention layer, not counting the masking operation, is composed of
>     * $6 N d^2$ FLOPs for the $Q,K,V$ linear layers
>     * $4N^2d$ FLOPs for the $QK$ dot product and the attention operation
>     * $3N^2h$ FLOPs for the softmax normalization.
>
> - Adding an edge attention bias per head (standard) adds another $+ N^2 h$ FLOPs
>
> - Adding color thus adds a further $+ 4N^2 (d-h)$ to the total cost.
>
> - The edge values are also responsible for another $+ N^2 d$ additions.
>
> Hence we come to a total of $6 N d^2+ 4N^2d + 4N^2h$ for the standard monochromatic attention and $6 N d^2+ 9N^2d$ for our full scheme (this is *per layer*).
>
> **Time:** We lack data from other graph transformers to perform an extensive comparison, but for similar-sized models from GraphGPS [Rampasek et al., 2022], and on similar hardware (a single A100 GPU) our method is just as fast on ZINC (20s/epoch), and around 30% slower on the other datasets (1500s/epoch vs 1100s/epoch for the *medium* models on PCQM4Mv2).
>
> We have added these numbers as Table 7 in the Appendix.
>
> **Memory cost**
> The model in itself has less parameters than the other graph transformers at comparable performance. However its GPU memory consumption can be quite high, as it requires $\mathcal{O}(N^2 d)$ floats for the full attention matrix, instead of $\mathcal{O}(N^2 h)$. This is manageable for small graphs (such as the molecular datasets in the benchmarks) but can be prohibitive for larger graphs.
>
> > 2. Lack the introduction of some baselines and metrics. Why you select these methods as baselines? Are they representative?
>
> We have added an explanation in the new version of the paper. We arrange the baselines in 3 broad categories: MPNNs, extended (or hierarchical) MPNNs, and graph transformers. We chose to include the best performing ones (to our knowledge) in each category, with an emphasis on the more comparable graph transformers. We deem they are representative of the field up until the first submission of this paper.
>
> > 3. Code is not published.
>
> The code is published but we did not give the link for sake of anonymity. We have now joined an anonymised version of the repository in the latest revision.
>
> > Add some statistics of the used datasets to show they have different scales from small to large.
>
> In the experiments section we give the number of graphs in each dataset, as well as their average number of nodes. When we write "medium" or "large" we refer to the number of graphs in the set. ZINC [1] is a popular medium-sized molecular benchmark, with 12K graphs, and PCQM4Mv2 [2] is the largest available molecular benchmark, with 3.7M graphs.
>
> > Introduce some baselines and metrics. For example, why you use MAE for some datasets and use accuracy for others?
>
> We use the standard evaluation metrics for these datasets (as introduced in their respective papers [1], [2]), to ensure fair comparison to other methods. We did not find justification for using one metric or another in the original papers though, maybe their authors are better suited to answer this question.
>
> We are happy to discuss if you have any further questions!
>
> [1]: Dwivedi et al. *Benchmarking Graph Neural Networks.* 2020
>
> [2]: Hu et al.  *OGB-LSC: A
> large-scale challenge for machine learning on graph.* 2021

---

### Review · Reviewer_F8E3 · 2023-07-12

**Summary Of Contributions:**

The paper studies the self-attention mechanism in graph transformers, which is said to be able to adapt to broader types of structured data such as 3D point cloud. The key idea is to extend the self-attention scores to the domain of feature channels. By integrating the graph structural information and original edge features, the proposed Chromatic Self-Attention CSA could independently modulate the feature channels and go beyond the local message-passing mechanism. Higher-order topological information such as a ring - type motif is incorporated to further enhance the expressiveness of CSA. Extensive experiments on real-world datasets demonstrate the effectiveness of the proposed method and achieve SOTA performance on the ZINC benchmark dataset.

**Audience:**

Yes

**Claims And Evidence:**

Yes

**Requested Changes:**

Please kindly refer to the Strengths and Weaknesses.

**Strengths And Weaknesses:**

Strengths:
* The paper studies the problem of the positional encoding in graph transformer, which is a novel and relevant topic of interest.
* The paper’s idea of extending the self-attention to the feature is interesting. The proposed method seems simple yet effective.
* The SOTA performance on two benchmark datasets validate the effectiveness of the proposed CSA method.

Weaknesses:
* I found several claims and notations that are difficult to follow. For example, the notations for the edge feature matrix are not consistent. It seems like $E_{ij}$ is a vector, but is $\mathbf{E}_{ij}$ from Equation (5) a matrix or vector as well? Meanwhile, the two different edge feature matrices $E^{att}$ and $E^{val}$ become matrices. I cannot also find the proper definition for both $E^{att}$ and $E^{val}$. Should $E^{val}$ be $E^{value}$? The authors are suggested to further proofread.
* Some claims seem vague in my opinion. For example, on page 6, the authors claim that, as I quote, “RWSE is commonly used as a node-wise SE but has to our knowledge never been used before as an RPE.” However, in the RWSE section, the authors show that Equation (12) is an extension of GraphiT. Does this mean that RWSE is already used in GraphiT as RPE but in a different form?
* The connection between edge features and node feature channels needs further discussion. Given the design of CSA are relatively intuitive, why the edge feature information does a great job in modulating the attention across node feature channels seems disconnected. Could the authors provide more insights into this design?
* Figure 3 is interesting. How about the attention scores of other channels aside from the shown three? Can the authors reach a consistent insight on what kind of structural pattern a certain channel prefers?

---

> ### Author Response · Authors · 2023-07-28
>
> Thank you very much for your thorough review, we hope the following points will answer your questions and comments.
>
> > I found several claims and notations that are difficult to follow. For example, the notations for the edge feature matrix are not consistent. It seems like $E_{ij}$ is a vector, but is $\mathbf{E_{ij}}$ from Equation (5) a matrix or vector as well? Meanwhile, the two different edge feature matrices $E^{att}$ and $E^{val}$ become matrices. I cannot also find the proper definition for both $E^{att}$ and $E^{val}$. Should $E^{val}$ be $E^{value}$? The authors are suggested to further proofread.
>
> Sorry for the confusion, we have updated the paper with unified notations and a definition for $E^{att}$ and $E^{val}$. In short, they are 2 learned linear transformations of the same initial edge features: $E^{att} = W^{att} E^{(0)}$ and $E^{val} = W^{val} E^{(0)}$. The weights $W^{att}$ and $W^{val}$ can be either layer-specific or shared across all CSA layers.
>
> > Some claims seem vague in my opinion. For example, on page 6, the authors claim that, as I quote, “RWSE is commonly used as a node-wise SE but has to our knowledge never been used before as an RPE.” However, in the RWSE section, the authors show that Equation (12) is an extension of GraphiT. Does this mean that RWSE is already used in GraphiT as RPE but in a different form?
>
> Indeed this deserves clarification. When we refer to RWSE we refer to its definition in [Dwivedi 2021], that is the stacking of the successive powers of the random walk matrix $ [RW^1, … , RW^k]$. In GraphiT a single power was chosen as a kernel function $K_r(i, j) = RW_{ij}^p$ , with no learnable parameters. So yes in some limited sense the random walk matrix has been used as an RPE.
>
> > The connection between edge features and node feature channels needs further discussion. Given the design of CSA are relatively intuitive, why the edge feature information does a great job in modulating the attention across node feature channels seems disconnected. Could the authors provide more insights into this design?
>
> Let us omit the softmax and set the $Q$, $K$ and $E^{val}$ matrices to 0 for clarity. In that setting the update rule is:
>
> - $$h_i \leftarrow h_i + \sum_{j=1}^{N_\text{nodes}} \exp(E^{att}_{ij}) \odot V_j.$$
>
> Each channel of the edge features $E_{ij}^{att}$ modulates the corresponding channel in the message from node $j$, $V_j = W_V h_j$. ($V_j$ and $E_{ij}^{att}$ are vectors of the same size $\in \mathbb{R}^d$)
>
> (Note that in this setting with no $QK$ dot product, our scheme is equivalent to standard multi-headed attention with $N_\text{heads} = N_\text{channels}$.)
>
> > Figure 3 is interesting. How about the attention scores of other channels aside from the shown three? Can the authors reach a consistent insight on what kind of structural pattern a certain channel prefers?
>
> The attention scores for all 16 channels are shown in Figure 4 of the appendix. Most of the channels seem to be variations on the 3 common themes shown in Figure 3, with some exceptions. This is to be expected as rings, pairwise distances and neighboring relations are all part of the initial edge features $E^{(0)}$. The channels of $E^{att}$ attribute different weightings to these components. The model will choose those weights during training, depending on the task.
>
> We are happy to further discuss if need be!

---

### Review · Reviewer_L9Qc · 2023-07-13

**Summary Of Contributions:**

This paper proposes CSA - a new "Chromatic" Self Attention mechanism. Here, the Chromatic sense stands for the ability to learn per channel weights that allow to dampen or highlight messages. The paper builds on recent advances in graph transformers such as GraphGPS. The authors show that the proposed CSA achieves state-of-the-art results on several benchmarks such as ZINC.

**Audience:**

Yes

**Broader Impact Concerns:**

I find no broader impact concerns.

**Claims And Evidence:**

Yes

**Requested Changes:**

Please see my notes in my main review.

**Strengths And Weaknesses:**

Pros:

The paper seems novel to me, and the idea is interesting. As stated by the authors, the proposed method can also be applied to other domains and this would be interesting to see how will it work.

The paper is well-written and easy to follow.

The results are promising and show a significant improvement compared to other methods.

Cons:

The citation format seems to be broken / not fitting the citation style of the paper. That is, when the authors cite a paper, in the middle of a sentence and then the name of the authors of the referenced papers show. I think this should be moved into parentheses, or change into numbered citations.

In page 3, "Node positional encoding", what is $r(\Lambda$) ?

There is a rather cryptic sentence in page 4, saying "On the other hand the heat diffusion or the PageRank kernels encode a notion of distance between nodes" that comes slightly out of context. Perhaps, the authors can expand on that.

The authors are missing a citation about shortest path encoding in GNNs "Shortest Path Networks for Graph Property Prediction".

Besides graph attention modules, another well performing approach on datasets like ZINC, is the use of subgraphs. I think that in order to provide a more broad comparison of existing methods, the authors should also add subgraph methods to their comparisons.

Regarding positional encoding methods with powers of the adjacency matrix or the random walk matrix, the authors should also discuss "SIGN: Scalable Inception Graph Neural Networks", "From Local to Global: Spectral-Inspired Graph Neural Networks", "Graph Positional Encoding via Random Feature Propagation".

In Section 3.5, the authors use $E^{(0)}$. Can the authors comment on the possibility of also updating the edge features and using them in deeper layers?

Have the authors tried to experiment with 3D point clouds as claimed? For instance, on point cloud classification?

---

> ### Author Response · Authors · 2023-07-28
>
> Thank you for your detailed review, we hope the following points will answer your questions and comments. If you have any other questions we are glad to discuss further.
>
> > The citation format seems to be broken / not fitting the citation style of the paper. That is, when the authors cite a paper, in the middle of a sentence and then the name of the authors of the referenced papers show. I think this should be moved into parentheses, or change into numbered citations.
>
> Thank you for pointing this out, we have moved everything into parantheses in the new version.
>
> > In page 3, "Node positional encoding", what is $r(\Lambda)$ ?
>
> We realize this part was not very clear, as it is the remainder of a longer digression that we chose to omit. We have simplified it in the new version.
>
> $r(\Lambda) \in \mathbb{R}^N$ was a function $r$ of the eigenvalues $\Lambda \in \mathbb{R}^N $ of the symmetric normalized Laplacian $L = I - D^{-\frac{1}{2}}AD^{-\frac{1}{2}}$.
>
> We were discussing in the previous paragraph how to design graph positional embeddings such that their scalar products $p_i \cdot p_j$ would encode some sort of spatial distance, akin to the cosine embeddings in NLP.
>
> We argue spectral embeddings $P = Diag(r(\Lambda)) U^T \in \mathbb{R}^{N\times N}$, where $U$ is the eigenvector matrix of $L$, would in theory be good candidates to do so. Indeed if $r(\lambda)=\sqrt{\lambda}$ then $P^T P = U Diag(\Lambda) U^T = L$, hence we get the property that the scalar products $p_i \cdot p_j$ replicate the adjacencies. If we chose $r(\lambda)=\sqrt{e^{-\beta\lambda}}$ then we have linearized the heat diffusion kernel $K_\beta = e^{-\beta L}$: $p_i \cdot p_j = K_\beta(i, j)$.
>
> As discussed further in the paragraph, these spectral embeddings have shortcomings limiting their performance in practice: converting $P$ into fixed-length vectors, resolving sign-flip and eigenvalue multiplicity ambiguities. These are amply discussed in the SignNet paper for example [Lim et al., 2022].
>
> > There is a rather cryptic sentence in page 4, saying "On the other hand the heat diffusion or the PageRank kernels encode a notion of distance between nodes" that comes slightly out of context. Perhaps, the authors can expand on that.
>
> If one chooses $K_r$ as the PageRank-$p$ kernel, then attention between two nodes $i$ and $j$ will be weighted by their "connectedness", ie the probability $K_r(i,j)$ of ending up at $j$ after $p$ random steps starting from node $i$. The PageRank and diffusion kernels encode a sort of *spatial* distance in graphs, by opposition to the comparison of local neighborhoods, obtained with nodewise RWSE positional encodings or substructure-counting for instance.
>
>
> > Besides graph attention modules, another well performing approach on datasets like ZINC, is the use of subgraphs. I think that in order to provide a more broad comparison of existing methods, the authors should also add subgraph methods to their comparisons.
>
> Indeed, we mention subgraph methods in the introduction, and have included the best performing one [Zhang et al., 2022] (to our knowledge) in the ZINC results. If there are other such references with good reasons to be included please indicate them.
>
> > Regarding positional encoding methods with powers of the adjacency matrix or the random walk matrix, the authors should also discuss "SIGN: Scalable Inception Graph Neural Networks", "From Local to Global: Spectral-Inspired Graph Neural Networks", "Graph Positional Encoding via Random Feature Propagation".
>
> From what we understand the first paper is a generalized MPNN that combines several graph operators (powers of the adjacency matrix for instance) for the message-passing operation, they do not build positional embeddings per se. The second one seems to be the same but with an extra normalization step.
>
> We have included the "Graph Positional Encoding via Random Feature Propagation" paper in our introduction, thank you for the reference.
>
> > In Section 3.5, the authors use $E^{(0)}$. Can the authors comment on the possibility of also updating the edge features and using them in deeper layers?
>
> In the model and code we give the option (edge parameter sharing = `False`) of using a different edge encoder at each layer l, resulting in layer-specific edge encodings $E(l)$. The differences in performance with sharing edge parameters across all layers are minor, depending on the task. Our results are computed with the best performing option for each task (hyperparameters available in the appendix).
>
> We have not thoroughly investigated updating $E(l+1)$ as a function of the previous layers, though it is an interesting idea. It is done in EGT [Hussain et al., 2022] for instance.
>
> > Have the authors tried to experiment with 3D point clouds as claimed? For instance, on point cloud classification?
>
> We have! We are currently working on adapting CSA to materials and crystallography, but it is too soon to include results in this paper.

---

### Decision · Action_Editors · 2023-09-11

**Recommendation:** Accept as is

**Comment:**

Reviewers find the proposed attention approach for Graph Transformers interesting and the experimental results strong. Main concerns were around clarity of some of the claims and computational cost. Authors have updated the draft addressing the clarity concerns, and additional results. One reviewer felt the paper needs more experiments to be convincing, but I think the paper has enough experiments to establish the usefulness of the proposed method. Overall I think the current draft has nice contributions and recommend acceptance.

**Audience:**

Yes, GNNs are an important and growing field.

**Claims And Evidence:**

All claims in the submission are accurate.